# Three-Step Description of Single-Pulse Formation of Laser-Induced Periodic Surface Structures on Metals

**DOI:** 10.3390/nano10091836

**Published:** 2020-09-14

**Authors:** Evgeny L. Gurevich, Yoann Levy, Nadezhda M. Bulgakova

**Affiliations:** 1Laser Center (LFM), University of Applied Sciences Münster, Stegerwaldstraße 39, 48565 Steinfurt, Germany; 2HiLASE Centre, Institute of Physics of the Czech Academy of Sciences, Za Radnicí 828, 2524 Dolní Břežany, Czech Republic; bulgakova@fzu.cz

**Keywords:** femtosecond laser, LIPSS, plasmons, two-temperature model, self-organization, three-step model

## Abstract

Two different scenarios are usually invoked in the formation of femtosecond Laser-Induced Periodic Surface Structures (LIPSS), either “self-organization” mechanisms or a purely “plasmonic” approach. In this paper, a three-step model of formation of single-laser-shot LIPSS is summarized. It is based on the periodic perturbation of the electronic temperature followed by an amplification, for given spatial periods, of the modulation in the lattice temperature and a final possible relocation by hydrodynamic instabilities. An analytical theory of the evolution of the temperature inhomogeneities is reported and supported by numerical calculations on the examples of three different metals: Al, Au, and Mo. The criteria of the possibility of hydrodynamic instabilities are also discussed.

## 1. Introduction

Laser-induced periodic surface structures (LIPSS), first observed by Birnbaum on semiconductor surfaces [1], can be found on different materials, such as metals, polymers [2], graphene [3,4], and wide-band gap dielectrics [5,6,7]. Although this phenomenon is nearly as old as laser material processing, its mechanisms are still under debate in the scientific community. The principal approaches for the analysis of LIPSS formation are based on two main mechanisms: (1) interference between the incident light and a surface scattered wave (SSW) with the latter to be often attributed to surface plasmons [8,9,10,11] and (2) hydrodynamic-like theories considering material reorganization in the molten surface layer [5,6,12]. The hydrodynamic-like theories involve different possible processes such as selective evaporation mediated by the local curvature of the surface described by the Kuramoto–Sivashinsky equations [5,6] or a hydrodynamic instability in the laser-induced melt pool [12,13,14,15].

In this paper, we focus on LIPSS induced by femtosecond laser pulses on metal surfaces and illustrate the results on the examples of three metals: Al, Mo, and Au. We demonstrate that LIPSS formation by single laser pulses is possible on incompletely smooth surface and propose a theoretical approach, which combines electrodynamic, hydrodynamic and two-temperature thermal conduction ideas. This paper summarizes and extends the main points of the proposed model, which were partly published in several papers [12,16,17,18], and provides a comparison of different materials as compared to our earlier work [17].

Possible extension of the proposed theory for semiconductors and dielectrics requires an analysis of ionization mechanisms [19,20] and is out of the scope of this paper. Multi-pulse exposure is also beyond the scope of the paper because we focus here on the experimental regimes when the first pulse already forms a periodic pattern on the surface. We note however that multi-pulse regimes imply a positive feedback from structures formed by previous pulses, which facilitates periodic pattern formation via three possible mechanisms: (1) surface absorption depends on the local angle of incidence, and thus the energy deposition will be periodically modulated over the surface [16,17]; (2) plasmon excitation is facilitated by the periodic surface structures [21,22,23,24]; and (3) conditions for development of hydrodynamic instabilities in the laser-induced melt depend on the melt depth [12,25] and can be influenced by the surface topology left by the previous pulses.

The paper is organized as follows. In Section 2, we briefly summarize our experimental findings, and Section 3 and Section 4 present the three-step model (TSM) of LIPSS formation. Its first step, the formation of a periodic profile of the electron temperature, is discussed in Section 3.1. The second step, imprinting the periodic pattern of the electron temperature on the lattice temperature profile with its following evolution, is presented in Section 3.2. The third step, which considers the development of hydrodynamic instabilities, is described in Section 4. The results are discussed and summarized in the Discussion.

## 2. Experimental Observation

A typical example of the LIPSS imprinted on Al surface after single-pulse exposure by femtosecond laser *Hurricane* (*Spectra Physics*) at normal beam incidence is presented in Figure 1. We note that the single-pulse LIPSS have periodicity close to laser wavelength [18] and are usually classified as low spatial frequency LIPSS (LSFL) [11]. We also mention that similar single-shot patterns were produced by pulses of a 1030 nm picosecond laser as well as by its second and third harmonics [18]. This fact demonstrates that the formation of the single-shot LIPSS does not depend on prepulses and nano- or picosecond pedestals, which are usually present in ultrashort pulses [26]. Indeed, as pulse-to-pedestal contrast is increasing for generated harmonics due to a nonlinear nature of the harmonic generation phenomenon, in such cases a low-peak-power nanosecond pedestal is not expected to premodify the sample surface before the incidence of the main pulse.

Compared to the LIPSS created in the regimes when several pulses are coupling to the same surface area [27], the structures generated by single pulses are less regular with numerous defects such as crossing of the ripples, bifurcations, and irregularities in orientation. However, the periodicity of the pattern in Figure 1 is clearly seen. Furthermore, the preferable orientation of the LIPSS is perpendicular to the laser light polarization as expected for metallic surfaces [11]. Such patterns, whose regularity coexists with some disorder, may suggest either a self-organization mechanism or that several types of the processes are involved in the LIPSS formation. In the following sections, we follow the second option and analyze, by employing the TSM, whether the structures are formed by a collaborative action of the surface-scattered waves and hydrodynamic self-organization processes.

## 3. Thermal Steps: Temperature Modulations along the Surface

LIPSS are often considered as a sort of direct laser interference patterning (DLIP) process. The DLIP implies that the surface is periodically ablated in the maxima of the interference pattern, which is formed by the interference of the incident wave and the SSW. For metals, Sipe et al. [9] have pointed out that surface plasmons can participate in the SSW formation, and this prediction agrees with spatial periods and orientation of LIPSS obtained in many works. However, measurements over a broad range of laser wavelengths attest to a deviation between predicted and observed pattern periods [18], suggesting that other mechanisms can also be involved in the pattern formation.

In the TSM [12,18], we invoke the modulation of the electron temperature $T_{e}$ as the first step of the LIPSS formation phenomenon. At a wavelengths of interest, we consider that the laser energy couples primarily with electrons of the target metal leading to an inhomogeneity in the absorbed energy distribution at the surface. Then, in order to modify the topography of the sample, this periodic modulation of the electron energy has to be imprinted on the lattice temperature $T_{l}$ that constitutes the second step in the TSM. Finally, the modulation of $T_{l}$ can induce a material relocation within the molten surface layer, resulting in the observed ripples, that is considered in the third step of the TSM. Each of these steps can facilitate or subdue formation of the periodic structures and influence the final LIPSS period. In this section, we present the first two steps of the model from analytical and numerical points of view.

### 3.1. Step 1: Modulation of the Electron Temperature

As mentioned above, the modulations in the electron temperature may originate from the redistribution of the laser energy at the surface of a not perfectly smooth sample. According to the Sipe theory [9], for metals irradiated in air, the spectrum of the spatial frequencies of modulations of the energy deposited at the surface peaks at a period close to the laser wavelength. It is associated with the interference of the incident wave and the SSW. More generally, any inhomogeneity such as beam profile hot spots, target irregularities or scratches can affect the spatial distribution of the energy deposition into the electronic subsystem of the metallic target.

Although it is not obvious whether the process can be represented by the coupling between a surface plasmon and the incident wave (see Appendix A), the plasmonic approach may be considered as a means to obtain a maximization of the possible modulation amplitude in the electromagnetic field intensity distribution at the surface of the metal. The amplitude of the SSW is hard to quantify theoretically without knowing the surface roughness spectrum, but it can be estimated from the published experimental data under the assumption that the spectrum of the surface roughness is broad. The SSW should not necessarily be a plasmonic wave but it can simply originate from the scattering of the incident wave on the surface roughness. However, following the suggestion of Sipe et al. that the plasmonic waves dominate on metallic surfaces [9], we will analyze the possible efficiency of the plasmon generation under the conditions corresponding to typical LIPSS experiments. We first consider the literature where the efficiency of plasmon excitation was measured under the following different conditions,

a nearly ideal case of monochromatic light with an angle of incidence corresponding to the surface plasmon excitation, which illuminates a nearly-ideal grating instead of a rough surface;a situation closer to laser ablation experiments, in which the light is monochromatic and where the surface satisfies the conditions for plasmon excitation but its roughness spectrum is broad; anda situation even closer to laser ablation experiment, in which spectrally-broad femtosecond laser pulses with varying incidence angle impinge on a nearly-ideal grating.

All of these three cases are more advantageous for the excitation of surface plasmons than the conditions at which the LIPSS are generated. From the experiments related to the situations listed above, the following corresponding information can be extracted.

The surface plasmon excitation on a high-quality metallic grating was first discovered by Robert Wood [21,23] in 1902, who observed a narrow dark line in the spectrum of a broad-band light source. According to his results, the intensity dropped by the factor of ten over a spectral range narrower than the distance between the sodium lines, i.e., within the spectral range Δλ<0.6 nm. This means that, under the conditions of surface plasmon resonance, at least 90% of the incident light excites collective oscillations of the electrons at the surface.Excitation of plasmons on gold ridges of rectangular profile positioned on a gold film was studied in [28]. Gratings with a different number of ridges were illuminated at normal incidence to the film surface by continuous monochromatic laser sources within a certain range of wavelengths. The maximum efficiency of plasmon excitation of 20% was demonstrated. Reducing the number of ridges to one decreased the plasmon excitation efficiency approximately by the factor of ten (down to 2–3%). We note that the single ridge case can be considered as a demonstration of a reduced plasmon excitation efficiency on a randomly rough surface.Experimental results on plasmon excitation on metal grating with femtosecond laser pulses at different angles of incidence were recently published by Miyaji et al. [29,30], who found that plasmons can be excited with broadband femtosecond laser pulses similar to ones used in our experiments. However, the best excitation efficiency of approximately 10% was measured only in a narrow range of angles of incidence approximately 25°. For incidence angles smaller than 20°, no excitation of plasmons was observed.

These data do not enable making of a quantitative estimation of the amplitude of excited plasmonic waves but give convincing arguments to assume that it is relatively small compared to the incident wave amplitude. We note that this is applicable to the case of a polished surface with a roughness in the sub-micrometer range exposed to single femtosecond laser pulses at the zero incidence angle. Then, the conclusion can be made that, if the interference between the incident wave and the SSW takes place, the amplitude of the intensity modulation along the surface for the resulting electromagnetic wave is very small compared to the intensity of the incident laser wave. We underline that it is due to the low efficiency of the plasmon excitation and, thus, large difference in the amplitudes of the interfering waves. Consequently, the periodic modulation of the temperature of electrons, which absorb the energy of the electromagnetic wave, is small as well.

In the following subsection, we will show that this small modulation of the electron temperature $T_{e}$ can result in a large modulation of the lattice temperature $T_{l}$.

### 3.2. Step 2: Lattice Temperature

For gaining insight into lattice heating due to the absorbed laser energy transfer from hot electrons, we apply the two-temperature model (TTM) [31]. We consider that the irradiation spot diameter is much larger than the period of LIPSS and assume that laser energy absorption is uniform along the grooves of the LIPSS (in *y* direction). Then, the dynamics of the energy coupling between electrons and lattice can be described in two dimensions, where the laser beam is impinging at normal incidence (*z* direction) on the material surface (*x* coordinate). In such configuration, the TTM governing equations can be written as
(1)$$\left\{\begin{array}{cc} c_{e}\left(T_{e}\right)\frac{\partial T_{e}}{\partial t}= & \frac{\partial }{\partial x}\kappa_{e}\left(T_{e},T_{l}\right)\frac{\partial T_{e}}{\partial x}+\frac{\partial }{\partial z}\kappa_{e}\left(T_{e},T_{l}\right)\frac{\partial T_{e}}{\partial z}-G\left(T_{e}-T_{l}\right)+I_{0}\left(x,t\right)\left(1-R\right)\alpha e^{-\alpha z}\\ \; & \\ c_{l}\frac{\partial T_{l}}{\partial t}= & \frac{\partial }{\partial x}\kappa_{l}\frac{\partial T_{l}}{\partial x}+\frac{\partial }{\partial z}\kappa_{l}\frac{\partial T_{l}}{\partial z}+G\left(T_{e}\right)\left(T_{e}-T_{l}\right) \end{array}\right.$$

Here, $c_{e}\left(T_{e}\right)=A_{e}T_{e}$ and $c_{l}$ are the electron and lattice heat capacities, respectively; $\kappa_{l}$ and $\kappa_{e}\left(T_{e},T_{l}\right)=\varkappa \frac{T_{e}}{T_{l}}$ are the lattice and electron thermal conductivities, respectively; and *G* is the electron-phonon coupling rate. The corresponding values employed for aluminum, gold, and molybdenum are summarized in Table 1. The absorption of the incident laser light is described by the last term in the first equation, where $I_{0}\left(x,t\right)$ is the intensity distribution on the bulk sample surface (z=0), and *R* and α are the reflection and absorption coefficients, respectively. In this work, we disregard the change of optical properties during the laser pulse irradiation for the sake of simplicity. It was shown, for the case of gold, that the temperature-dependent absorption can provide an additional positive feedback mechanism in the growth of the amplitude of the electron temperature modulations [17,32]. In the following, we therefore consider absorbed intensities and fluences.

The following analysis is focused on the time after the end of the laser pulse, so that $I_{0}\left(x,t\right)=0$ and the last term in the first equation of the system (Equation 1) vanishes. We assume that, during the laser pulse action, a periodically modulated electron temperature profile appears for the reasons analyzed above. Thus, we introduce the modulation as an harmonic single mode perturbation of the average dynamics of the electron temperature: $T_{e}=\left(T_{e}^{0}\left(t\right)+{\tilde{T}}_{e}\left(t\right)e^{ikx}\right)e^{-\xi z}$. Here, $T_{e}^{0}\left(t\right)$ denotes the average electron temperature on the sample surface, ξ is a characteristic in-depth scale of the temperature profile, and k=2π/Λ and ${\tilde{T}}_{e}\left(t\right)$ are the wave number and the amplitude of the temperature modulation of the period Λ, respectively. The characteristic in-depth scale ξ can be obtained from one-dimensional numerically-calculated temperature profiles for both the electrons and the lattice. We determine it as the inverse value of the distance (along *z*), at which the temperature drops from its maximum at the surface to the 1/e level in the bulk of the sample. The $T_{e}$ and $T_{l}$ spatial profiles dynamically change, and thus the corresponding ξ values evolve in time as shown in Figure 2 for three studied metals. However, during the time interval between the end of the laser pulse $t_{0}$ and the characteristic coupling time $t_{c}$ (the time at which $T_{e}\left(t\right)$ and $T_{l}\left(t\right)$ dependences recorded for the sample surface cross [31]), the ξ values remain of the same order of magnitude. This holds also for the relative difference between the ξ values for electrons and lattice, which stays below 50% during the whole considered time interval (note some difference between $\xi (T_{e})$ and $\xi (T_{l})$ even after $t_{c}$ that is explained by a fast heat conduction of electrons so that the electron-lattice thermalization below the metal surface prolongs beyond $t_{c}$). For simplicity in the following analysis, we use for each material the same constant value of ξ for the electrons and lattice.

Due to the coupling between the electron and lattice subsystems, the $T_{e}$ modulated profile can transfer modulation to the $T_{l}$ profile. Thus, we consider the lattice temperature profile as $T_{l}=\left(T_{l}^{0}\left(t\right)+{\tilde{T}}_{l}\left(t\right)e^{ikx}\right)e^{-\xi z}$.

Assuming that the perturbation develops on a time scale shorter than that of the evolution of the average temperature, i.e., $\frac{d{\tilde{T}}_{e}\left(t\right)}{dt}\gg \frac{dT_{e}^{0}\left(t\right)}{dt}$, the system of partial differential equations, see Equation (Equation 1), can be written as a system of ordinary differential equations for the amplitudes of the surface temperature modulation ${\tilde{T}}_{e}$ and ${\tilde{T}}_{l}$. The stability of this system of equations can be verified via its rewriting to the matrix form $\dot{\vec{T}}=A\vec{T}$, by introducing the vector $\vec{T}={\left({\tilde{T}}_{e},{\tilde{T}}_{l}\right)}^{T}$ and the matrix A (see in [17]) that writes as
$$A=\left(\begin{array}{ccc} \frac{2\varkappa \xi^{2}}{A_{e}T_{l}^{0}}-\frac{\varkappa k^{2}}{A_{e}T_{l}^{0}}-\frac{G}{A_{e}T_{e}^{0}} & \; & -\frac{\varkappa T_{e}^{0}\xi^{2}}{A_{e}{T_{l}^{0}}^{2}}+\frac{G}{A_{e}T_{e}^{0}}\\ \frac{G}{c_{l}} & \; & \frac{k_{l}\xi^{2}}{c_{l}}-\frac{k_{l}k^{2}}{c_{l}}-\frac{G}{c_{l}} \end{array}\right)=\left(\begin{array}{cc} a_{11} & a_{12}\\ a_{21} & a_{22} \end{array}\right)$$

The evolution of a small perturbation of the temperature can be analyzed according to the signs of eigenvalues of the matrix A. However, instead of finding the eigenvalues, one can apply the trace-determinant criterion [38]: the matrix A is unstable (i.e., any small perturbation of the surface temperature grows) if at least one of the following conditions is fulfilled.

Trace of the matrix T>0Determinant of the matrix D<0

Notice that the matrix coefficients depend on material constants, which in their turn can depend on the average surface temperature (which evolves slowly on the time scale of $10^{-11}-10^{-10}$ s), wave number of the instability *k*, and the characteristic heating depth 1/ξ. For the three metals under study, the initial periodic modulations with periods corresponding to wavelengths of the visible part of the spectrum should continue growing after the end of the laser pulse. Jumping slightly ahead, we note that high relative amplitudes of the lattice temperature modulation $a_{l}\left(t\right)=T_{l}^{\max}-T_{l}^{\min}$ ($T_{l}^{\max}$ and $T_{l}^{\min}$ are the maximum and minimum lattice temperatures along the surface) can be achieved, see Figures 4–6. The mechanism of $a_{l}\left(t\right)$ amplification can be understood as a nonlinear effect connected with the electron thermal diffusivity $D_{e}=\kappa_{e}\left(T_{e},T_{l}\right)/c_{e}\left(T_{e}\right)=\varkappa /A_{e}T_{l}$ as shown in Figure 3. The laser energy absorbed by free electrons relaxes through two dissipation channels: the electron–lattice coupling and the electron thermal diffusivity. The electron–lattice coupling results in emerging the $T_{l}$ maxima in the regions of the $T_{e}$ maxima. This decreases heat diffusion from these hotter regions due to reduced $D_{e}$. As a consequence, the confined electron energy couples to the lattice in these specific regions, thus providing a positive feedback loop responsible for amplification of the lattice temperature modulation [17]. From numerical simulations, it follows that, for some pattern periods, the lattice temperature remains substantially modulated for at least 0.5 ns. This time is large enough for this modulation to guide a hydrodynamic instability in the melt with formation of a surface relief, which in its turn can be imprinted to the surface upon rapid resolidification (Figure 4, Figure 5 and Figure 6).

### 3.3. Numerical Modeling

We complement the results of step 2 of the TSM by numerical simulations of the temperature evolution at the surfaces of the studied metals upon irradiation by an ultrashort laser pulse. The simulations support the validity of the second step in the TSM, enable a more accurate description of the problem (in-depth profile, nonlinearity), and can take into account other physical aspects such as a dynamic change of optical properties upon heating of the electrons during the laser pulse and temperature-dependent thermophysical properties (not addressed here, however).

The 2D numerical code is based on the equations of the two-temperature model (Equation 1). For all materials, we consider a single laser pulse, Gaussian in time, with a pulse duration of $\tau_{\mathrm{las}}=130\;\mathrm{fs}$ (full width at half maximum). The simulations start at t=0 and the maximum of the Gaussian pulse intensity is reached at $t=+2\tau_{\mathrm{las}}$. Following previous works [16,17,39], we focus on the evolution of the amplitude of the temperature modulations at the surface of the materials. The modulations arise after introduction of a single mode harmonic modulation in the laser pulse intensity profile along the *x* direction. The spatial part of the intensity has the form $I_{0}\left(x,t\right)=I_{0}\left(t\right)\;\left(1+A\cos\left(kx\right)\right)$, where A is the amplitude of the modulation of spatial period Λ. The amplitude considered in the present work is A=0.05, corresponding to an intensity peak-to-peak variation of 10%. We note that recently Terekhin et al. [40] proposed a way of accounting for the modulated energy deposition by direct simulations of the interference between a SSW and the incident laser pulse. However, for the sake of simplicity, in this study we restrict ourselves to the use of a harmonic, single mode perturbation. The modes tested here numerically are summarized in Table 2. Similarly to the analytical model, the uniform distribution of the absorbed energy in the *y* direction is assumed, and thus we address the case of the ripples oriented along the *y* axis.

For direct comparison with the analytical model, the numerical calculations are based on the same thermophysical properties (summarized in Table 1). For each metal, the laser pulse fluence was adjusted in order to reach surface temperatures close to but below their respective melting temperatures $T_{m}$. Therefore, the following absorbed fluences have been employed. For gold, $F_{0}=110\;J\;{\mathrm{cm}}^{-2}$; for aluminum, $F_{0}=13.5\;J\;{\mathrm{cm}}^{-2}$; and for Mo, $F_{0}=80\;J\;{\mathrm{cm}}^{-2}$.

As mentioned previously, due to the perturbation introduced in the source term, a modulation appears along the *x* direction in the electronic (first step in the TSM) and lattice temperature profiles (second step). Lateral (i.e., along the surface) thermal diffusion, however, contributes to heal the inhomogeneities at the surface. This is particularly the case for the stronger temperature gradients, i.e., for the smallest spatial periods. Because the thermal diffusivity of the lattice is several orders of magnitude smaller than that of the electrons, the modulation imprinted in the lattice is not expected to be smoothed out via lattice thermal conduction. It rather experiences coupling with the electronic subsystem that is a much more efficient channel of heat evacuation. There is consequently a relatively complex interplay between thermal diffusivity of the electrons, lateral heat flow (and thus spatial period), in-depth heat flow, and the electron-lattice coupling.

The differences $T_{e,l}^{\max}\left(t\right)-T_{e,l}^{\min}\left(t\right)$ between the minima and the maxima for the temperature profiles of the electrons and the lattice along the surface provide the amplitudes of the modulations $a_{e}$ and $a_{l}$, respectively. The evolution of $a_{e}$ and $a_{l}$ values is presented in Figure 4, Figure 5 and Figure 6 for gold, aluminum, and molybdenum. Each curve corresponds to a numerical simulation involving a single spatial frequency *k* in the modulation of the laser intensity profile $I_{0}\left(x,t\right)$, corresponding to the periods ranging from 100 nm to 8 μm. For the three metals under investigation, the calculations show the strong increase of the amplitude, $a_{e}\left(t\right)$, of the perturbation in the electron temperature during the first part of the laser pulse, demonstrating the deposition of the modulated laser energy to the electrons. Due to the electron–phonon coupling, the lattice temperature also becomes modulated. It is notable that, within the time interval between $t_{0}$ and $t_{c}$, the $a_{l}$ values of some spatial periods grow, whereas those of some others, of small periods, decrease. The sets of simulated spatial periods experiencing a global decrease during this time interval are different for Au, Al, and Mo, revealing a non-trivial entanglement of the lateral and in-depth thermal diffusion with the electron–lattice coupling.

### 3.4. On the Evolution of the Lattice Temperature Modulations

The Trace-Determinant criterion is a simple tool to analyze the ability of the initial temperature modulation to grow, but it cannot predict the characteristic timescales for this growth. This can be done by calculating the eigenvalues of the matrix A and their corresponding real and imaginary parts $\lambda_{1,2}=\lambda_{1,2}^{r}+i\lambda_{1,2}^{i}$. The real part of the largest eigenvalue gives the characteristic time scale for the growth of the lattice temperature modulation, while the imaginary part (if any) indicates the ability of this modulation to oscillate in time (provided that the frequency of such oscillations is high enough to be observed before the temperature modulation $a_{l}$ decays). The eigenvalues can be calculated as solutions of the equation
$$Det\left(A-\lambda I\right)=Det\left(\begin{array}{cc} a_{11}-\lambda & a_{12}\\ a_{21} & a_{22}-\lambda \end{array}\right)=0.$$

The eigenvalues for Al, Au, and Mo obtained with this formula for different wave vectors *k* are shown in Figure 7, left. It has been found that they have both real and imaginary parts. The eigenvalues for gold slightly differ from that reported in [17], where we used a different method for estimation of the thermal depth 1/ξ and the contribution of the lattice thermal conductivity. Here, for all metals (Table 1), the latter is taken as 1% of the electron thermal conductivity. To compare with the numerical results, we define the growth rate of the modulation amplitude, obtained by the simulations, as $\Gamma =\left(a_{l}\left(t_{c}\right)-a_{l}\left(t_{0}\right)\right)/\left(a_{l}\left(t_{0}\right)\left(t_{c}-t_{0}\right)\right)$. Its values for three metals are presented in Figure 7, right.

The simplified analytical model shows that the growth rates for gold are in a good agreement with the numerical calculations [17]. The eigenvalues for Al and Mo have large imaginary parts. This implies that the amplitude of the temperature modulation can oscillate. However, numerical simulations show that the growth rates for Al and Mo are constant and very close to each other. This can be explained by the fact that an arbitrary selection of the incident fluence in the simulations intrinsically limits the growth rate for Al, which has a lower melting point than Mo. Although according to simulations the growth rates for molybdenum are comparable with those for aluminum (Figure 7), the amplitude of its temperature modulation (Figure 6) is higher than in Al, which can be explained by the longer coupling time and the higher melting temperature. This underlines an important role of thermophysical properties of materials in the modulation growth rates that calls for further studies.

## 4. Relocation Step: Hydrodynamic Instabilities

In the previous section, we discussed how the lattice temperature modulation grows with time due to a periodic modulation of the electron temperature. In this section, we analyze how a periodically modulated lattice temperature can cause the LIPSS formation, i.e., produce a periodic relief with the height-to-depth amplitude of the order of $10^{-7}$ m on the metal surface. We assume that the reorganization of atoms on the surface can be influenced by the temperature that leads to imprinting the periodic lattice temperature modulation to the sample surface in the form of LIPSS. First we find criteria, which can help in selecting possible instabilities; afterwards, we discuss the possibilities for different instabilities to develop.

We note that, in this third step of the TSM, the hydrodynamic relocation of the molten metal is not the only possible mechanism leading to a change in the topography of the sample surface. For example, a modulated ablation may contribute to the LIPSS formation. We emphasize that the modulations in the lattice temperature appear not only on the surface, but also beneath the surface. Therefore, at larger fluences, when ablation of the material develops in the forms of vaporization, phase explosion, spallation, or formation of subsurface voids [6,41,42,43,44], a modulated lattice temperature will naturally lead to a modulated surface relief.

### 4.1. Criteria of Hydrodynamic Instabilities Development

Hydrodynamic instabilities manifest themselves in a reorganization of the molten material on the surface. These processes start if any kind of driving force for such a reorganization overbalances the friction forces. The ratio between the driving and friction terms in the Navier–Stokes equation can be represented as characteristic numbers serving as the criteria to assess whether a certain type of instability can develop or not. For example, the Marangoni number $M=\frac{\sigma_{T}^{\prime }\Delta Td}{D\eta }$ characterizes the importance of the forces related to the surface tension, while the Rayleigh number $Ra=\frac{\beta \Delta Td^{3}\rho \ddot{z}}{D\eta }$ evaluates the possible contribution of the inertial forces. These numbers can be calculated for a given set of parameters such as the depth of the molten layer *d*, surface tension σ and its temperature derivative $\sigma_{T}^{\prime }$, difference between the melt temperature at the top and at the bottom of the laser-molten layer ΔT, acceleration of the surface $\ddot{z}$, thermal diffusivity *D*, volumetric thermal expansion coefficient β, dynamic viscosity η, and some others. The analysis of the characteristic numbers allows a fast and easy conclusion about the roles of different mechanisms upon the LIPSS formation as it was systematically shown for fused silica [7]. If a characteristic number exceeds some critical value, the corresponding instability can develop.The disadvantage of this simplified approach is in the assumption that the above-mentioned parameters are constants or slowly changing in space and time. If this assumption is not valid, a more complex analysis is required (for details see, e.g., in [45]). The advantage is that it renders unnecessary the solution of the Navier–Stokes equations, which are extremely hard to solve [46].As shown in numerical simulations [16] and in Figure 4, Figure 5 and Figure 6, the surface temperature modulation may remain substantial during a characteristic time $t_{\delta }∼10^{-10}-10^{-9}\;$s. This gives the second criterion that the growth rate of the instability must be $\gamma ≳t_{\delta }^{-1}∼10^{10}-10^{9}$ s${}^{-1}$.Hydrodynamic instabilities can develop in a certain range of spatial periods where each period has a positive growth rate. The maximal γ corresponds to the so-called fastest growing mode (fgm). The period of the fastest growing mode $\lambda_{fgm}$ defines the observed period resulting from the instability if it has enough time to develop. The latter may not always be the case due to rapid solidification after the laser pulse. Moreover, the linear theory of the instabilities is formally valid only for small amplitudes of the surface profiles. In the other words, it is difficult to make a universal statement about the relation of the LIPSS period to $\lambda_{fgm}$.When an instability starts to develop, the amplitudes of the different spectral components of the surface profile h(k) depend on time *t* as $h\left(k\right)=a_{0}\left(k\right)\times e^{\gamma (k)t}$. Here, k=2π/Λ and $a_{0}\left(k\right)$ is the initial spectrum of the amplitudes, which is established by, e.g., the SSW as discussed in the previous sections. The spatial mode *k* corresponding to the maximum of the h(k) can change in time: if γ(k)t≪1, the exponential factor can be neglected and the observed spectrum of the LIPSS will be close to the initial spectrum of SSW, $a_{0}\left(k\right)$. If the instability has more time to develop before the resolidification occurs, the period of the LIPSS will be closer to the fastest growing mode $\lambda_{fgm}$ because the exponent will dominate in this case. This consideration explains the experimentally observed wavelength dependence of picosecond LIPSS on copper [18].

### 4.2. Analysis of the Instabilities

The question “what kind of hydrodynamic forces dominates in the redistribution of the melt upon the laser ablation?” is still open. However, based on the criteria discussed above, one can identify probable and rule out improbable candidates for the dominating physical mechanisms of the LIPSS formation in this relocation stage.

The **Marangoni instability** is the most frequently mentioned mechanism of the material reorganization upon LIPSS formation. It develops if the Marangoni number M is larger than a critical value $M_{c}∼10^{2}$ [47]. Analytical estimations [24] and numerical analysis [48] show that the Marangoni number for laser-ablated metals is two orders of magnitude lower than the minimal required value. This makes the growth of this particular instability impossible in ultrafast laser ablation experiments. A periodic surface modulation of the temperature due to, e.g., SSW can facilitate the Marangoni instability, but any plausible temperature modulation is not enough to make this mechanism realistic [12].

The **Rayleigh–Taylor instability** is a more probable candidate for the dominating hydrodynamic flow mechanism. It was evidenced that this instability induces periodic structures upon laser ablation of spherical and cylindrical targets used for inertial confinement fusion [49,50]. It can develop if the Rayleigh number Ra exceeds the critical value of $Ra_{c}∼10^{3}$. In the traditional presentation of the Rayleigh–Taylor instability, the acceleration of the surface is considered to be driven by gravity, which is, however, negligible in our case. Using $\ddot{z}=g\approx 9.8$ m/s${}^{2}$ yields the Ra value ~12 orders of magnitude lower than $Ra_{c}$. However, in the case of the irradiation by ultrashort laser pulses, the melt surface is not an inertial system, and thus its acceleration should be reconsidered.

As discussed in [12], the surface of the melt accelerates after the heating by the laser pulse, but the acceleration direction is different in different phases of the process. First, it is directed from the melt toward the air, which cannot cause the instability. However, after the melt cools down, the surface is decelerated, which supports the development of the Rayleigh–Taylor instability. Moreover, recoil force due to evaporation is also directed from the surface toward the bulk, and thus the associated acceleration also maintains the instability. The destabilizing acceleration due to these two mechanisms can be estimated as $\ddot{z}∼10^{12}$–10${}^{13}$ m/s${}^{2}$, which allows fulfilling the criterion of the instability in the case of ultrashort laser action on materials. Shih et al. [15] proposed another deceleration mechanism. According to their molecular dynamics simulations of metals ablated in water, the development of an instability was assigned to the deceleration of the ablation products by the environment.

## 5. Discussion

Although the necessity of combining the electrodynamic and hydrodynamic approaches of the LIPSS formation was pointed out by many authors [13,14,41], the existing models of this process look like a competition between these two approaches. However, as shown in this paper, their combination can fill the gaps, which a separate approach is unable to explain self-consistently. Here, we combine these two approaches in three steps, which can be easily realized for metals:the periodic modulation of the **electron temperature** is assumed from the coupling of the incident laser pulse with the surface roughness;the periodic modulation of the **lattice temperature** is induced by the relaxation between electrons and phonons; andthe formation of the **surface profile** is due to melt relocation, which is induced by the hydrodynamic instabilities in the thin melt layer, influenced by the temperature modulation.

The last step of the TSM is coupled to the previous one as the initial perturbation of the instability is induced by the modulated lattice temperature. We notice, however, that other mechanisms of the surface profile formation like spallation and evaporation [6,42,44] or formation of subsurface voids [41,43] are also controlled by the lattice temperature and can be included as the third step of the TSM.

All three steps of the TSM (see Figure 8) can affect the LIPSS period. Indeed, the initial modulation of the electron temperature has a certain spectral width with the maximum, which does not necessarily correspond to the maximum growth rate of the lattice temperature perturbations. In this case, the modulation period of the lattice temperature will be formed in competition between the period of the SSW interference pattern and the growth rates shown in Figure 7. In the same way, the dispersion curve of the hydrodynamic instability can again shift the observed LIPSS period. Finally, we stress that the hydrodynamic instability does not necessarily require any initial temperature modulation and that the surface relief can grow out of noise. This can happen if the roughness of the surface is too low or if the polarization and/or spectral characteristics of the incident light are not defined [51].

The extension of the TSM on dielectrics and semiconductors is possible but requires careful consideration of ionization processes, which is out of the scope of this paper. The first step of the model can be additionally facilitated by the positive feedback induced by temperature-dependent optical properties of metals [27].

## Figures and Tables

**Figure 1 nanomaterials-10-01836-f001:**
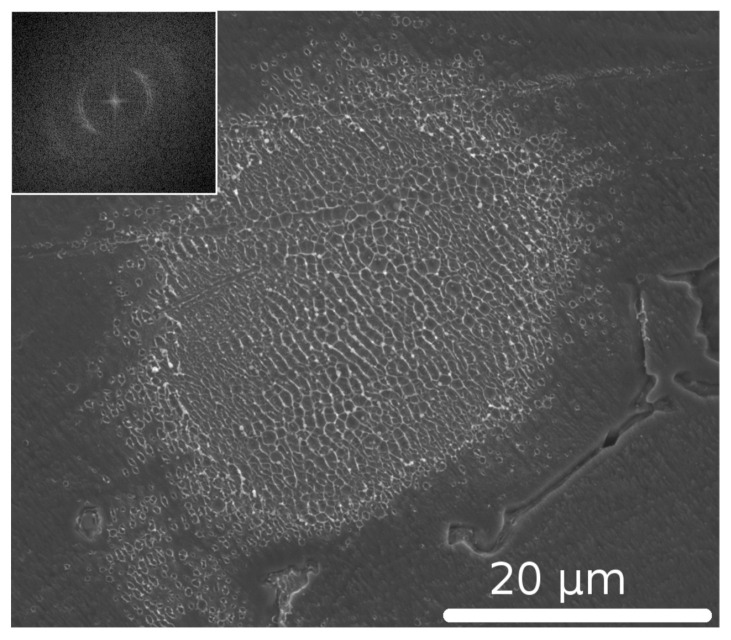
SEM image of LIPSS on Al formed by a single ultrashort laser pulse at laser wavelength λ=800 nm with pulse duration $\tau \approx 10^{-13}$ s and pulse energy $E_{p}\approx 25\;\mu$J. Irradiation spot diameter was approximately 30 μm. Inset: Fast Fourier transform analysis of the central part of the irradiation spot showing the emergence of a peculiar spatial period of the pattern and a relatively broad distribution of ripples orientation.

**Figure 2 nanomaterials-10-01836-f002:**
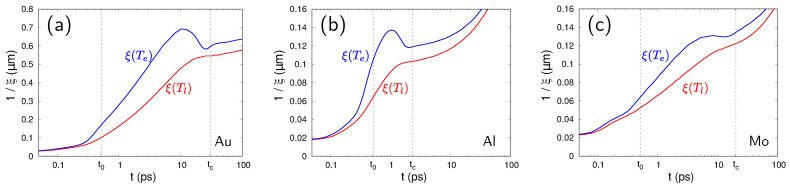
Evolution of the characteristic in-depth lengths (1/ξ) of the temperature profiles for lattice (red) and electrons (blue) calculated for Au (**a**), Al (**b**), and Mo (**c**). Vertical dashed lines mark the end of the laser pulse and the electron–phonon coupling time determined here as the time when the $T_{e}\left(t\right)$ and $T_{l}\left(t\right)$ curves cross (∼30ps for Au, ∼2ps for Al, and ∼20ps for Mo).

**Figure 3 nanomaterials-10-01836-f003:**
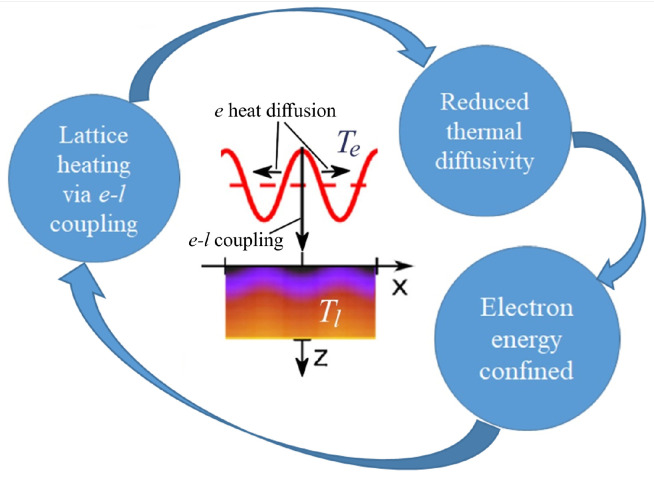
Schematics of the feedback loop responsible for amplification of the lattice temperature modulation during the electron–lattice coupling stage. See text for the details.

**Figure 4 nanomaterials-10-01836-f004:**
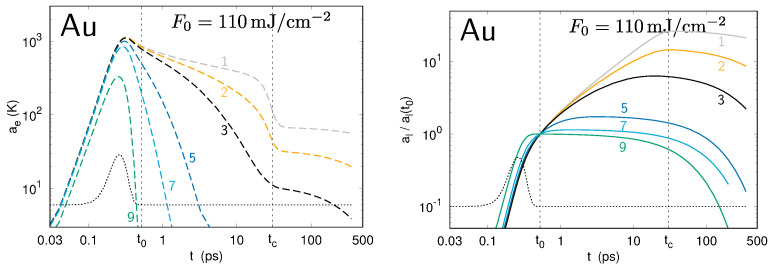
Evolution of the amplitudes $a_{e}$ (**left**) and $a_{l}$ (**right**) of the perturbation in the electron and lattice temperatures, respectively, at the surface of gold upon irradiation by a single laser pulse. Each curve corresponds to a different spatial period Λ in the modulation of the transverse intensity (see numbering in Table 2). For the lattice temperature, the amplitude is normalized to its value at $t_{0}$: we plot $a_{l}\left(t\right)/a_{l}\left(t_{0}\right)$.

**Figure 5 nanomaterials-10-01836-f005:**
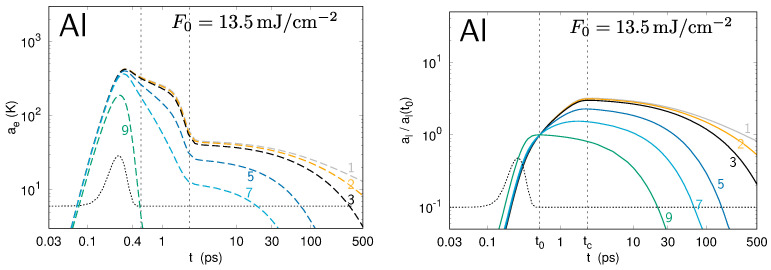
The same as in Figure 4 for aluminum.

**Figure 6 nanomaterials-10-01836-f006:**
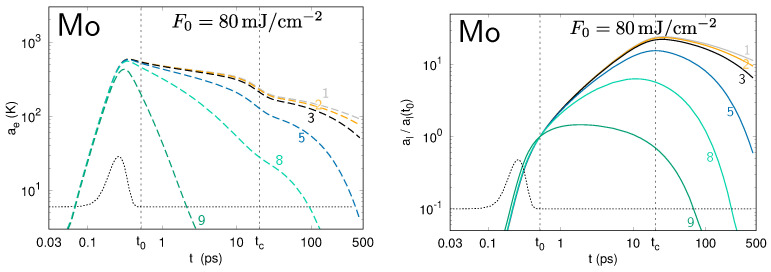
The same as in Figure 4 and Figure 5 for molybdenum.

**Figure 7 nanomaterials-10-01836-f007:**
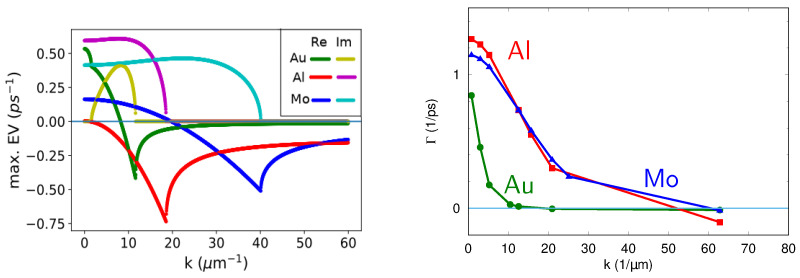
(**Left**) real and imaginary parts of the maximal eigenvalues (EV) of the matrix A, corresponding to the growth rates estimated in the analytical model. Calculations are made for Al ($\xi =9.0\times 10^{6}$ m${}^{-1}$), Mo ($\xi =1.56\times 10^{7}$ m${}^{-1}$), and Au ($\xi =5.9\times 10^{6}$ m${}^{-1}$). (**Right**) numerically calculated growth rates for the time interval between $t_{0}$ and $t_{c}$.

**Figure 8 nanomaterials-10-01836-f008:**
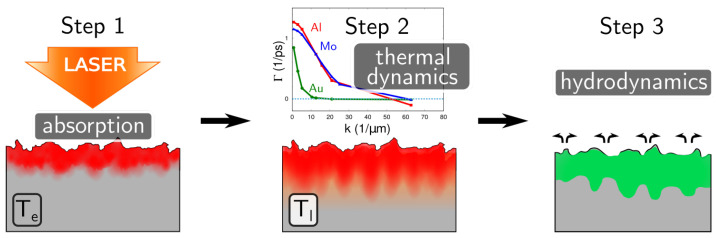
Schematic representation of the TSM.

**Table 1 nanomaterials-10-01836-t001:** Parameters used in the stability analysis. $T_{m}$ denotes the melting temperature.

Parameter	Units	Gold	Aluminum	Molybdenum
$A_{e}$	$J\;m^{-3}\;K^{-2}$	71 [33]	95.3 [34] (fit)	350 [33]
κ	$W\;m^{-1}\;K^{-1}$	318 [33]	237 [35]	135 [33]
*G*	$W\;m^{-3}\;K^{-1}$	$2.1\times \;10^{16}$ [33]	$3.1\times \;10^{17}$ [36]	$1.3\times \;10^{17}$ [33]
$c_{l}$	$J\;m^{-3}\;K^{-1}$	$2.5\times \;10^{6}$ [33]	$2.43\times \;10^{6}$ [35]	$2.8\times \;10^{6}$ [33]
$k_{l}$	$W\;m^{-1}\;K^{-1}$	3.18 (1% of κ)	2.37 (1% of κ)	1.35 (1% of κ)
$T_{m}$	K	1337 [33]	933 [35]	2897 [33]
$\alpha^{-1}$	nm	12.4 [37]	7.52 [37]	19.0 [37]

**Table 2 nanomaterials-10-01836-t002:** Summary of the spatial frequencies/periods of the initial modulation considered in the numerical calculations.

Numbering	1	2	3	4	5	6	7	8	9
*k* (1/μm)	0.785	2.99	5.24	10.5	12.6	15.7	20.9	25.1	62.8
Λ (μm)	8.00	2.10	1.20	0.600	0.500	0.400	0.300	0.250	0.100

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
