# Peer review of "Three-Step Description of Single-Pulse Formation of Laser-Induced Periodic Surface Structures on Metals"

_nanomaterials, 2020, doi:10.3390/nano10091836_

Round 1
Reviewer 1 Report
This manuscript provides valuable insight in the complex problem of the formation of periodic features as a result of laser irradiation.
The observation and discussion of the single pulse effect in particular will make in my opinion this article a very interesting read to the relevant scientific community.
The manuscript addresses the interesting issue of the origin of periodic surface features caused by single-pulse laser irradiation of metallic surfaces. This is done by building a comprehensive model that takes into account the hydrodynamic instabilities of the laser-melted surface along with the electromagnetic interaction to explain the spontaneous periodic pattern formation on the surface of metals with a wide range of physical properties. Although the analysis presented in this paper is restricted to metals it also constitutes a good starting point for the understanding of this effects on the surface of semiconductors and dielectrics.
The most important feature of this paper is the attempt to explain the formation of periodic features after single pulse irradiation, which minimizes the contribution of positive feedback originating from diffraction, and therefore addressing fundamental aspects of the periodic pattern formation.
The paper is clearly written, well balanced and coherent. In my opinion it will be an important addition to the field of laser-mater interaction and therefore merits publication.
Reviewer 2 Report
This paper represents a worthwhile attempt to formulate a rational explanation For LIPSS formation, but considering the three step nature of energy transfer (laser pulse -> electron gas, e-gas -> lattice, lattice -> motion of molten metal).
However, in order for the authors' ideas and conclusions to reach the reader, significant improvements in the logic and clarity of presentation must be made.
In order for the model to be clear, the authors are advised to seek the following modifications:
- equations presented are useful, but the origins of amplification need to be explained in a straightforward basic way, by identifying non-linearity of the relationship between T_e and T_l mathematically, and illustrating them graphically
- discussion of Marangoni and Rayleigh numbers must be made clear: authors list relevant parameters without giving dimensionless forms, which are only introduced later, making it awkward for the reader to follow
- the final Figure does not provide a good illustration / explanation - improve
Also
acceleration of the surface g - this is later identified as gravity constant; which is it
thermal expansion beta - which thermal expansion? volumetric?
Best provide nomenclature!
And please, make a further effort to make the paper better - it can go a long way, if thought through thoroughly!
293 coefficient b
Round 2
Reviewer 2 Report
Good effort to revise and improve -
use of materials acceleration instead of gravity acceleration may deserve more detailed thought and discussion - but probably best outside the present manuscript.